# Precise Positioning Method of Moving Laser Point Cloud in Shield Tunnel Based on Bolt Hole Extraction

**Changqi Ji, Haili Sun \*, Ruofei Zhong, Jincheng Li and Yulong Han**

Beijing Advanced Innovation Center for Imaging Theory and Technology, Key Laboratory of 3D Information Acquisition and Application, MOE, College of Resource Environment and Tourism, Academy for Multidisciplinary Studies, Capital Normal University, Beijing 100048, China
\* Correspondence: sunhaili@cnu.edu.cn

**Abstract:** Mobile laser scanning technology used for deformation detection of shield tunnel is usually two-dimensional, which is expanded into three-dimensional (3D) through mileage, resulting in low positioning accuracy. This study proposes a 3D laser point cloud positioning method that is divided into rings in the mileage direction and blocks in the ring direction to improve the positional accuracy for shield tunnels. First, the cylindrical tunnel wall is expanded into a plane and the bolt holes are extracted using the self-adaptive parameter adjustment cloth simulation filter (CSF) algorithm combined with a density-based spatial clustering of applications with noise (DBSCAN) algorithm. Second, the mean-shift algorithm is used to obtain the center point of the bolt hole, and a model is designed to recognize the center point of different splicing blocks. Finally, the center point is combined with the standard straight-line equation to fit the straight-line positioning seam, achieving an accurate ring and block segmentation of a shield tunnel as a 3D laser point cloud. The proposed method is compared with existing methods to verify its feasibility and high accuracy using the seams located by the measured tunnel point cloud data and in the measured point cloud. The average differences between the circumferential seams positioned using the proposed method and those in the point cloud at the left waist, vault, and right waist were 3, 4, and 5 mm, respectively, and the average difference between the longitudinal seams was 3.4 mm The proposed research method provides important technical and theoretical support for tunnel safety monitoring and detection.

**Keywords:** precise positioning; 3D laser point cloud; shield tunnel; CSF; monitoring; measurements

## 1. Introduction

In recent years, mobile laser scanning technology has been widely used in shield tunnel deformation detection owing to its advantages of high efficiency and precision. However, the obtained data are comprised of a two-dimensional cross-section point cloud, which needs to be expanded into three dimensions in combination with mileage. Generally, the accuracy of mileage measurement is low, resulting in poor positioning accuracy in the direction of mileage. At the same time, circular positioning has always been an urgent problem to be solved.

China's urban rail transit has developed rapidly in recent years. A total of 50 cities in mainland China opened 283 urban rail transit lines with a total length of 9206.8 km by the end of 2021. Among them, the length of subway lines is 7209.7 km, accounting for 78.3% of urban rail transit [1]. Due to the influence of environmental and human factors, structural deformation and damage will occur in the operation process of modern structures, which will lead to significant decline and even destruction of structural performance over time [2]. After a tunnel is excavated, it will undergo a certain degree of structural deformation owing to the influence of various factors, such as changes in the soil around the tunnel, which affects the safety of the tunnel itself and the passage of vehicles [3–5]. Therefore, it is very important to perform a regular inspection of all tunnels. There are higher requirements

for the efficiency of detection due to the short window available owing to normal subway operations and maintenance. Traditional tunnel deformation monitoring methods, such as total station and convergence meter, mostly analyze the deformation by measuring the tunnel section on site. Although the accuracy of these methods is high, the efficiency is low [6–11]. In addition, the traditional methods for modern structure detection require complex instruments and a large number of sensors installed on the main body of the structure, which brings great challenges to deformation and damage detection [12–14]. Mobile laser scanning technology can obtain high-precision and high-density tunnel laser point clouds in a short time and has gradually become an important means of deformation detection for rail transit tunnels. As a result, the tunnel point cloud data processing method based on mobile laser scanning technology has become a research hotspot [15–22]. At present, the representative mobile tunnel laser inspection systems include the GRP IMS5000 mobile scanning system developed by AMBERG in Switzerland [23], TS3 tunnel scanning system developed by SPACETEC in Germany [24], SiTrack: One mobile track scanning system developed by the Leica Company [25], Orbit Mobile 3D Laser Measurement System (rMMS) of the Wuhan Hanning Rail Transit Technology Co., Ltd. of China [26], and Mobile Tunnel Measurement System developed by Capital Normal University [21,27]. All these systems can quickly acquire high-density point clouds of tunnels by integrating sensors, such as laser scanners, odometers or inertial measurement units.

The shield method is the main construction type of subway tunnels. The shield tunnel lining is generally assembled from various types of segments, including a capping block (KT), two adjacent blocks (BT), and an indefinite number of standard blocks (AT). Each segment is connected by a certain number of bolts; therefore, the most notable feature of the shield tunnel is the large number of evenly distributed bolt holes and seams on the lining [28–30]. There are two kinds of seams: The longitudinal seam between the pipe pieces in the ring and the circumferential seam between the ring pieces. A mobile laser scanning system generally integrates laser scanners, odometers, and manual push or electric detection trolleys. The acquired data are comprised of a two-dimensional cross-section point cloud, which is generally expanded into three dimensions through mileage. The point cloud of the shield tunnel must be divided into rings through the positioning of the annular seams to accomplish the positioning in the mileage direction. The positioning in the ring is achieved by positioning the longitudinal seam for inner-ring segmentation. Therefore, locating the seam accurately and efficiently is a key problem to be solved in the data processing of a mobile laser scanning shield tunnel.

Chen et al. [31] proposed a circumferential seam recognition method based on the feature of distance difference. According to the distance difference between the discrete points of the circumferential seam and the tunnel surface, a certain threshold is set to identify the circumferential seam, but this method cannot identify the occluded circumferential seam. There are relatively few studies on annular seam recognition in point cloud, the majority of which are based on 3D laser scanning grayscale images of tunnels. Du et al. [32] proposed a gradient statistical circumferential seam recognition method based on image intensity, which selects point cloud data within a certain angular interval from each section to generate a grayscale image of the tunnel lining, calculates and counts the gradient value of the image pixel along the width (mileage) direction of the grayscale image, and finally determines the circumferential seam using the peak detection method. Liu et al. [33] based their method on the mathematical morphological characteristics of a grayscale image of the circumferential seam of a shield tunnel segment. The authors used the image sliding window method, histogram equalization, scaling, threshold judgment, and other methods to automatically identify the circumferential seam. Cheng et al. [34] used a grayscale orthophoto of a tunnel lining surface as the sample data, You Only Look Once V3 to train the samples, and the obtained weight file to identify the circumferential seams. The aforementioned methods were all based on the grayscale images of the tunnel lining. Therefore, the recognition accuracy of the circumferential seam was greatly affected

by factors, such as the material, appendages, roughness of the inner wall of the tunnel, and illumination inside the tunnel.

Sun et al. [35] proposed a method based on the Canny operator edge detection and Hough transform line detection algorithms to identify longitudinal seams. First, they selected the point cloud within the range of the capping block in the circumferential direction to generate a grayscale image, then used the Canny operator to calculate the boundary of the image and expanded it. Finally, the Hough transform detection algorithm was used to detect the longitudinal seams on both sides of the capping block. Moreover, this method relied on a grayscale image of the tunnel lining, and the recognition accuracy was impacted by the influence of factors, such as the material, appendages, roughness of the inner wall of the tunnel, and lighting inside the tunnel.

This study proposes a method to locate the seam according to the distribution characteristics of bolt holes that is based on the regular structural characteristics of a circular shield tunnel. This method is based on a 3D point cloud and eliminates the influence of factors, such as the material, accessories, roughness of the inner wall of the tunnel, and internal light of the tunnel. At the same time, the positioning accuracy of the seam is improved considering that the circumferential seam is at a certain angle with the center line of the tunnel. The ring and block segmentation of shield tunnel 3D point cloud data achieves an accurate positioning effect, verifying that the positioning accuracy of the proposed method is reliable through the measured data.

## 2. Materials and Methods

This study combines the geometric position characteristics of the target in the scene to separate the tunnel track, remove appendages in the scanning data, and facilitate the processing of tunnel wall data. Combined with the cylindrical projection method, the cylindrical tunnel wall projection is transformed into a planar tunnel wall. On this basis, the bolt holes are extracted using the improved cloth simulation filter (CSF) algorithm [36], and their noise is removed using a density-based spatial clustering of applications with the noise (DBSCAN) clustering algorithm. In addition, the mean-shift clustering algorithm is used on each cluster of bolt hole point clouds to extract the center point. Subsequently, the centers generated by clustering individual non-bolt hole point clouds are eliminated by identifying the bolt hole centers belonging to various blocks. Generally, there is only one capping block in a ring for a shield tunnel, and the number and layout of bolt holes on the capping block also have certain particularity. The bolt hole center of the subordinate capping block is identified using the uniqueness of the capping block in each ring and the particularity of its bolt hole layout. Then, the bolt hole center that is subordinate to other segments is determined according to the corresponding relationship between other bolt holes in the ring and capping block to further eliminate the noise. Finally, the position of the seam is located according to the center of the bolt hole. The positional relationship between the point and straight line is used to identify the point clouds belonging to different splicing blocks. The blocks are represented by different colors to achieve the precise ring and block segmentation of the 3D laser point cloud of the shield tunnel. The implementation process of the proposed method is shown in Figure 1.

### 2.1. Point Cloud Preprocessing

The point cloud preprocessing of the shield tunnel mainly includes the separation of the tunnel track, removal of the appendages inside the tunnel, and projection expansion of the cylindrical surface of the tunnel wall.

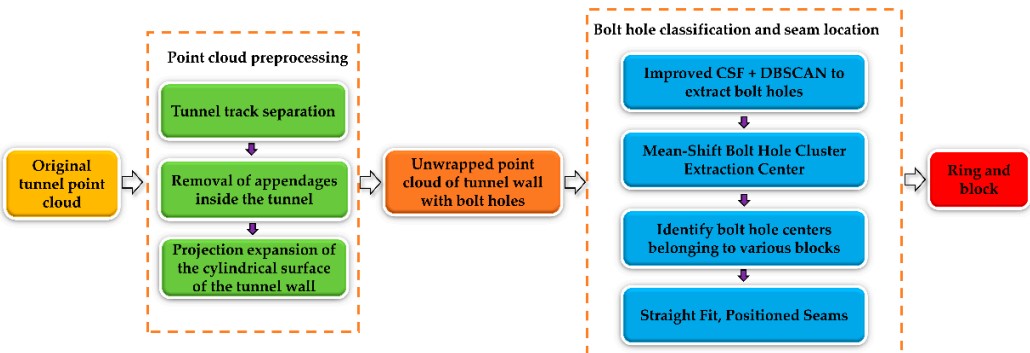

**Figure 1.** Flowchart of bolt hole extraction and seam location of shield tunnel.

### 2.1.1. Tunnel Track Separation

The original tunnel point cloud of the mobile measurement method is the section line, which contains the track information. The tunnel track can be separated in combination with the scanning angle. Traversing all sections can strip the tunnel data from the original data and delete useless information. The section coordinate system uses the scanner center as the coordinate origin. The $z$-axis is perpendicular to the track surface and points upward, and the $x$-axis is perpendicular to the $z$-axis and points to the right, as shown in Figure 2. The section center point $C(x_0, z_0)$ is obtained from the scanner center $O\ (0,0)$ plus translation parameters. The vector $\overrightarrow{CP}$ is formed by connecting the center $C(x_0, z_0)$ and point $P(x_i, z_i)$ on the tunnel section. Setting the positive direction of the $x$-axis as $0°$, the angle $\beta$ is formed by the vector $\overrightarrow{CP}$ and positive direction of the $x$-axis to separate the track and tunnel. A schematic of the section point cloud track tunnel separation was collected using a mobile tunnel measurement hardware system developed by the Capital Normal University, as shown in Figure 2 [29].

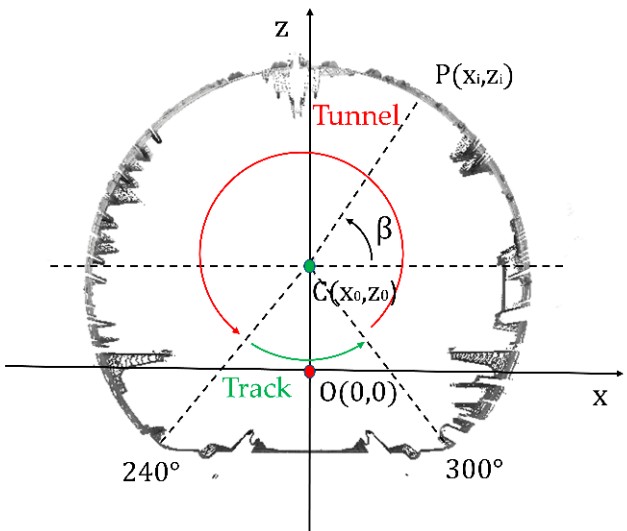

**Figure 2.** Schematic of the track and tunnel separation.

### 2.1.2. Removal of Appendages inside the Tunnel

A shield tunnel section can be considered as an ellipse, which is close to a standard circle [37]. The iterative ellipse fitting method was used to remove the point cloud of the appendages inside the tunnel at the section point, and the random sample consensus algorithm was used to estimate the parameters during ellipse fitting [17,38–42]. First, the initial ellipse fitting was carried out for the tunnel section with noise. Second, the initial parameters of the ellipse were calculated, such as the coefficient of the ellipse equation, coordinates of the center of the circle, long and short semi-axes, and inclination angle of

the long semi-axis. Third, the shortest distance $d_i$ from the section point to the ellipse was calculated. Finally, the distance set $d\ \{d_1, d_2, ..., d_n\}$ was formed and the mean $d_{mean}$ and standard deviation $\sigma$ of set d were calculated, which are respectively expressed as [43]:

$$d_{mean} = \frac{\sum_{i=1}^{n} d_i}{n} \tag{1}$$

$$\sigma = \sqrt{\frac{\sum_{i=1}^{n}(d_i - d_{mean})^2}{n-1}} \tag{2}$$

where $d_i$ is the distance from each section point to the elliptic curve, $d_{mean}$ is the distance mean value, and $\sigma$ is the standard deviation of distance. It was stipulated that the cross-section points were not tunnel wall points when $|d_i - d_{mean}| > 2\sigma$. After continuous iterations, the point cloud of the appendages inside the tunnel could be eliminated and the cross-section line was fitted.

Certain conditions needed to be added to ensure that the bolt hole points were not rejected. The initial parameters of the ellipse (geometric center $(X_0, Y_0)$, major semi-axis $aa$, minor semi-axis $bb$, and major semi-axis inclination angle $\theta$) were obtained using ellipse fitting. The focal radius c and coordinates of the two focal points $A(X_1, Y_1)$, $B(X_2, Y_2)$ can be respectively calculated using the following:

$$c = \sqrt{aa^2 - bb^2} \tag{3}$$

$$\begin{cases} X_1 = X_0 + c\cos\theta \\ Y_1 = Y_0 + c\cos\theta \end{cases} \tag{4}$$

$$\begin{cases} X_2 = X_0 - c\cos\theta \\ Y_2 = Y_0 - c\cos\theta \end{cases} \tag{5}$$

The sum $l_i$ of the distances from each point $U_i\ (x_i, y_i)$ of the section to the two focal points A, B was calculated, the distance set $l\{l_1, l_2, ..., l_n\}$ was obtained, and the distance $L$ from any point $P_i\ (x_p, y_p)$ on the ellipse to the two foci was calculated, as shown in Figure 3a. Moreover, the section point was the bolt hole point when $l_i > L$. Therefore, the bolt hole points were reserved. The tunnel wall point cloud with bolt holes removed from the tunnel interior appendages is shown in Figure 3b.

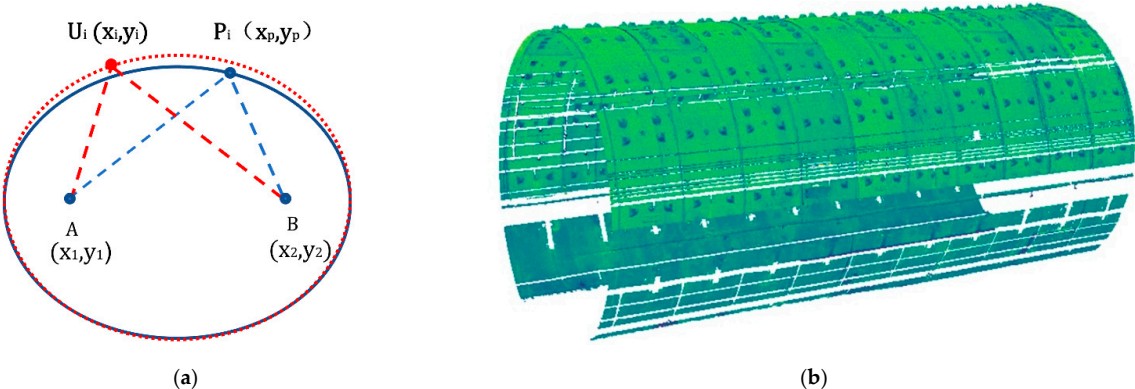

(a)

(b)

**Figure 3.** Effect drawings of: (**a**) Bolt hole point determination and (**b**) internal tunnel accessories removal.

### 2.1.3. Projection Expansion of the Point Cloud of the Cylindrical Tunnel Wall

This study draws on the idea of airborne 3D point cloud filtering to extract bolt holes. The overall structure of the tunnel point cloud is cylindrical. The cylindrical tunnel point cloud needs to be flattened before the bolt hole extraction is performed on the tunnel point cloud. Therefore, projection was used to simulate the spatial relationship between the ground and non-ground point clouds in the airborne 3D point cloud.

A cross-sectional coordinate system was established using the geometric center obtained from ellipse fitting as the origin, vertical upward ray as the positive direction of the $z$-axis, and $x$-axis perpendicular to the $z$-axis pointing to the right, as shown in Figure 4a. Taking each section as a unit, the angle $\theta$ between the $z$-axis and section points on the left and right sides of the $z$-axis were calculated using:

$$\theta = \begin{cases} \arctan\left(\frac{x_i - x_0}{z_i - z_0}\right) & x_i \geq 0, z_i \geq 0 \\ \arctan\left(\frac{x_i - x_0}{z_i - z_0}\right) & x_i \leq 0, z_i \geq 0 \\ -\frac{\pi}{2} - \arctan\left(\frac{z_i - z_0}{x_i - x_0}\right) & x_i \leq 0, z_i \leq 0 \\ \frac{\pi}{2} - \arctan\left(\frac{z_i - z_0}{x_i - x_0}\right) & x_i \geq 0, z_i \leq 0 \end{cases} \tag{6}$$

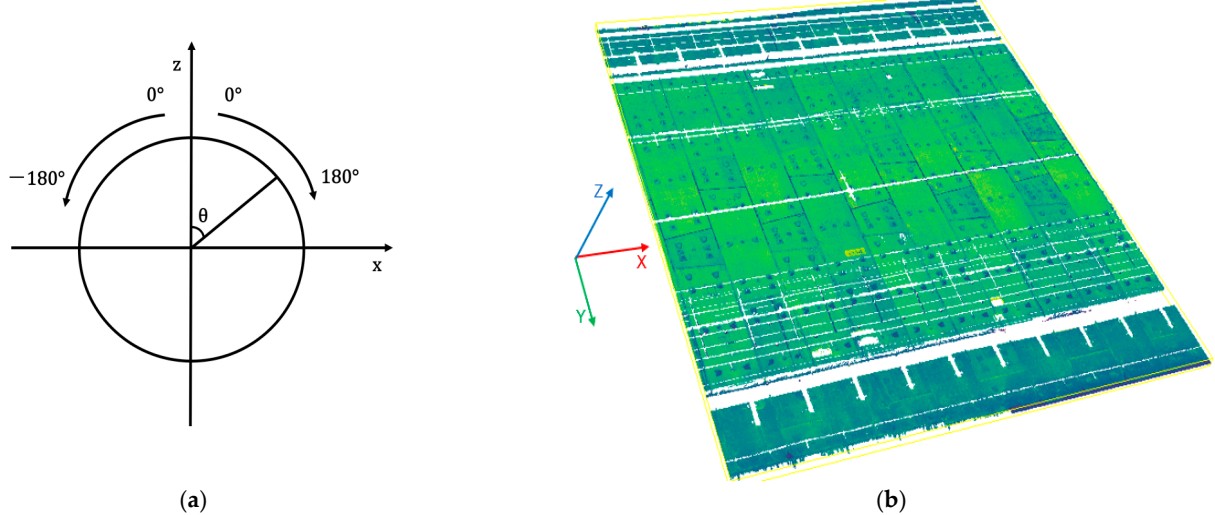

**(a)** **(b)**

**Figure 4.** (**a**) Tunnel projection and (**b**) tunnel wall expansion effect.

The arc length $L$ of each point was combined with the design radius $R$ of the tunnel segment and calculated with respect to the positive direction of the $z$-axis according to the arc length formula $L = R \times \theta$. Then, it was recorded as the $Y$ coordinate value of each point after projection. The $Z$ coordinate value of each point after projection was the distance from each point before projection to the geometric center of the ellipse. Simultaneously, the $X$ coordinate value of the point cloud after the projection of each section was recorded directly as the mileage value of the corresponding section ($y$-coordinate before projection). Seven fields were set to save the data in the forms of $x$, $y$, $z$, and $i$ before projection and $X$, $Y$, and $Z$ after projection to facilitate the restoration of the original state of the point cloud, as listed in Table 1. The effect after the tunnel was unfolded as shown in Figure 4b.

**Table 1.** Post-projection data structure.

| Pre-Projection $x$ (m) | Pre-Projection $y$ (m) | Pre-Projection $z$ (m) | Intensity $i$ | $X$ after Projection (m) | $Y$ after Projection (m) | $Z$ after Projection (m) |
|---|---|---|---|---|---|---|
| 1.830 | 19,155.287 | −1.087 | 126 | 19,155.287 | 6.535 | 2.746 |
| 1.835 | 19,155.287 | −1.081 | 118 | 19,155.287 | 6.527 | 2.745 |
| 1.842 | 19,155.287 | −1.076 | 115 | 19,155.287 | 6.518 | 2.746 |

### 2.2. Extraction and Classification of Bolt Hole Center Point

The improved CSF algorithm with an adaptive parameter adjustment and DBSCAN algorithm were used to extract the bolt hole point cloud for the expanded tunnel wall point cloud [44]. Then, the mean-shift algorithm was used to cluster the bolt hole point cloud to

generate the center point, and a bolt hole recognition model was designed to identify the bolt hole center belonging to various blocks.

### 2.2.1. Bolt Hole Point Cloud Extraction

A bolt hole is a structure protruding outward in space relative to the tunnel wall, which can be extracted by integrating airborne point cloud filtering. This study was inspired by the CSF in airborne laser point cloud filtering, where the shield tunnel point cloud containing bolt holes is compared to the airborne ground point cloud, the bolt hole point cloud is compared to ground objects, and the bolt hole point cloud extraction is carried out in combination with the cloth filtering principle. The principle of the CSF algorithm is to first flip the original airborne point cloud data up and down, then project the flipped point cloud and simulated cloth nodes on the same horizontal plane. This process finds the corresponding point of each cloth node in the point cloud and records the height of the corresponding point before the projection. The current height of the node is compared in each iteration, and iterations are repeated until the maximum number is reached or the node elevation changes are small enough to terminate the process. Finally, the distance between the laser point and cloth node is calculated. The point is marked as a non-ground point when the distance is greater than a certain threshold; otherwise, it is marked as a ground point. Based on the CSF wave to extract the bolt hole point cloud, the DBSCAN algorithm was used to remove the noise in the bolt hole point cloud.

The CSF algorithm has three adjustable parameters: Threshold, resolution, and number of iterations. The threshold is used to establish the point at a certain height from the cloth node as the ground point. The resolution defines the size of the cloth grid. If the setting value is small, the calculation will be large, and if the setting value is large, the properties of many point clouds will be misjudged. In principle, this is similar to point spacing or two to three times the point spacing. The number of iterations is the condition used to control the program stop. The improved CSF and DBSCAN algorithms are outlined below.

1.  Modification of Cloth Simulation
    (1)  Set the "correct position" of the cloth particles to the initial position, which is above the maximum elevation value after the flipped point cloud of the tunnel wall.
    (2)  Modify the classification threshold to $hcc = \mid (max + a) - ave \mid + b$; where $max$ is the highest value of the point cloud elevation, $a$ is the difference between the cloth node and highest value, $ave$ is the average value of the point cloud elevation, and $b$ is the compensation parameter used to adjust the effect of the separation between the tunnel wall and bolt hole.
    (3)  The cloth resolution is set to 3 times the average value of the radial and circumferential point spacing.

2.  The Process of Extracting Bolt Holes Using the Improved CSF Algorithm with the Self-Adaptive Parameter Adjustment Combined with the DBSCAN Algorithm
    (1)  Invert the tunnel Light Detection and Ranging (LiDAR) point cloud.
    (2)  The mesh resolution of the initial cloth mesh is set to 3 times the point spacing, and the cloth position is set above the highest point.
    (3)  Obtain the classification threshold $hcc$.
    (4)  Iterate the loop and calculate the height difference between the LiDAR point cloud and cloth particles.
    (5)  Distinguish the tunnel wall and bolt hole points. If the distance between the LiDAR point and cloth particle is greater than the threshold $hcc$, it is considered as a bolt hole point; otherwise, it is considered as a non-bolt hole point.

The extracted bolt hole point cloud contained noise since the height difference between the bolt hole points and some seam points was small. The two parameters of the DBSCAN algorithm, *Eps* (radius of the adjacent area) and *minPts* (number of points contained in the adjacent area at least), were set to appropriate values to remove the noise according to the

difference between the bolt hole point cloud and noise density. The effect of extracting bolt holes using CSF and DBSCAN algorithms is shown in Figure 5.

(**a**)                                                                                           (**b**)

**Figure 5.** Effect drawing of extracted bolt holes. (**a**) Tunnel lining gray image and (**b**) extraction of bolt hole point cloud.

### 2.2.2. Bolt Hole Clustering to Obtain the Center

The mean-shift [45] clustering algorithm was used to cluster each bolt hole point cloud into one class and calculate the center point of the bolt hole. The key function of the mean-shift algorithm is to calculate the drift vector of the center point according to the transformation of the data density in the region of interest, in order to move the center for the next iteration until it reaches the maximum density (the center point remains unchanged). This can be performed from each data point, and the number of times the data appears in the region of interest can be counted in this process. Bandwidth is a key parameter of mean-shift, which is used to determine the size of the region of interest. Here, it was set as the radius of the maximum circumscribed circle of the bolt hole. The clustering results are shown in Figure 6.

### 2.2.3. Identifying Bolt Hole Centers Subordinate to Various Blocks

The bolt hole center identification template is defined according to the distribution characteristics of bolt holes. The template identifies the bolt hole centers belonging to the capping, adjacent, and standard blocks. Subsequently, the non-bolt hole centers generated using mean-shift clustering are removed. The bolt hole center of the capping block was first identified owing to the apparent distribution characteristics of its six bolt holes. The identification template of the bolt hole center of the capping block was defined according to its design data, as shown in Figure 7. Point $C$ $(x_0, y_0)$ was selected as the center of the circle, and the other n (n $\geq$ 6) points are in the area with $R$ as the radius. Subsequently, it was determined whether the other n-1 points (except point $C$) were within the given five rectangles.

The centers of the bolt holes belonging to the adjacent and standard blocks were determined according to the position relative to point C in the capping block. Then, the center points of all bolt holes after clustering were traversed, which divided them into the center points of the bolt holes belonging to the capping, adjacent, and standard blocks. Each block was assigned a different color to distinguish them. The final classification results, in which the noise was further removed, are shown in Figure 8.

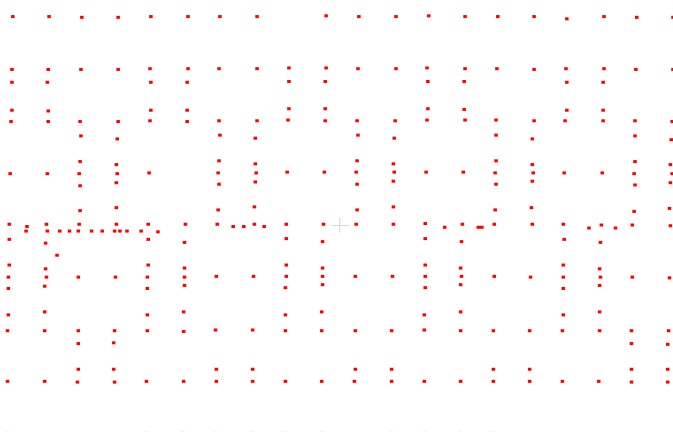

**Figure 6.** Mean-shift clustering results.

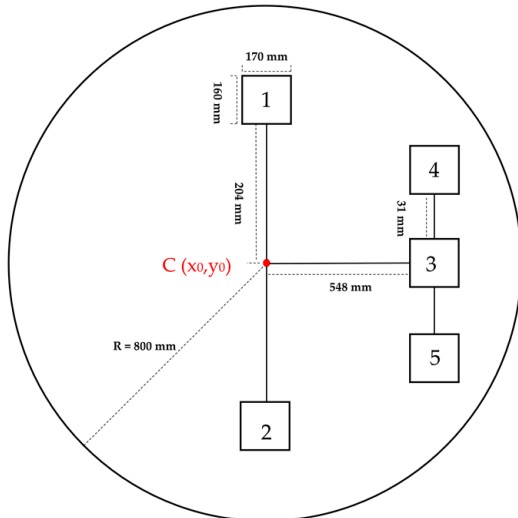

**Figure 7.** Capping block bolt hole identification template. Serial numbers 1–5 indicate the positions of other bolt holes in the capping block.

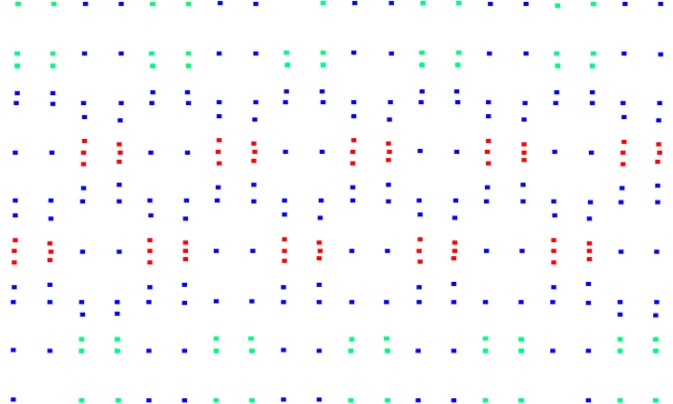

**Figure 8.** Classification results of bolt hole centers. Red, blue, and green dots represent the capped, adjacent, and standard blocks, respectively.

### 2.3. Seam Positioning and Ring Block Segmentation

The straight-line fitting method was adopted to attain the location of the seam. The center points (A1, A2, B1, B2, C1, C2, D1, and D2) were formed by the clustering of the bolt holes connecting the capping block, as shown in Figure 9a. The left and right adjacent blocks were interpolated to obtain points A, B, C, and D. Points (A,B) and (C,D) were used to obtain straight lines 1 and 2, respectively. Then, the lines were used to locate the longitudinal seams on the left and right sides of the capping block.

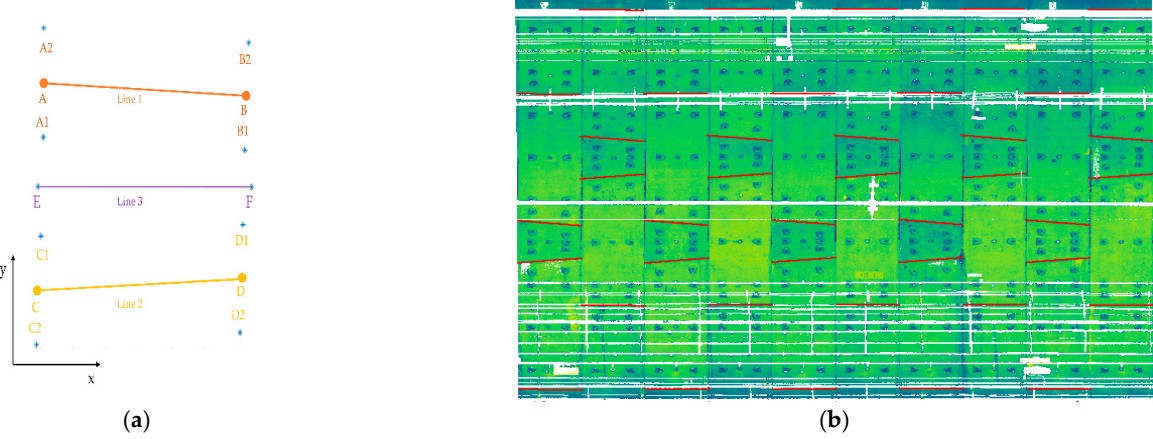

**Figure 9.** (**a**) Straight-line fitting effect and (**b**) transverse seam positioning effect of cross seam.

The center points (a1, a2, b1, b2, c1, c2, d1, d2, e1, e2, f1, f2, g1, g2, h1, and h2) were formed by the clustering of the bolt holes connecting two adjacent rings and were interpolated to obtain a, b, c, d, e, f, g, and h, respectively, as shown in Figure 10a. The straight-line equation was used to fit and locate the circumferential seam. The positioning effect of the circumferential seam is shown in Figure 10b.

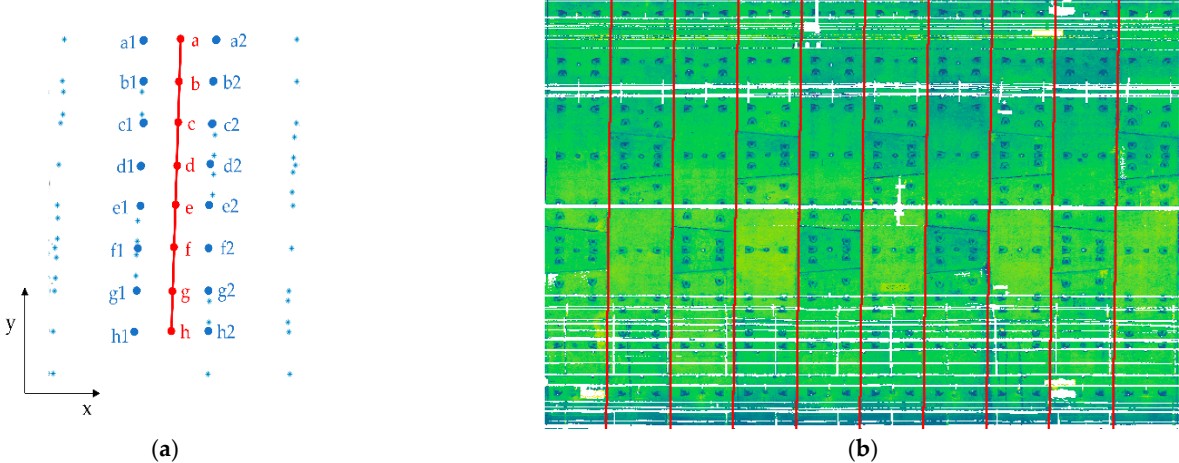

**Figure 10.** (**a**) Straight-line fitting effect and (**b**) positioning effect of circumferential seams between rings.

The 3D laser point cloud of the shield tunnel was divided into rings and blocks to achieve precise positioning according to the positioned seam and relationship between the point and straight line. The positioning effects in the flat and original states are shown in Figure 11a,b, respectively. Red, blue, and green colors represent the capped, adjacent, and standard blocks, respectively.

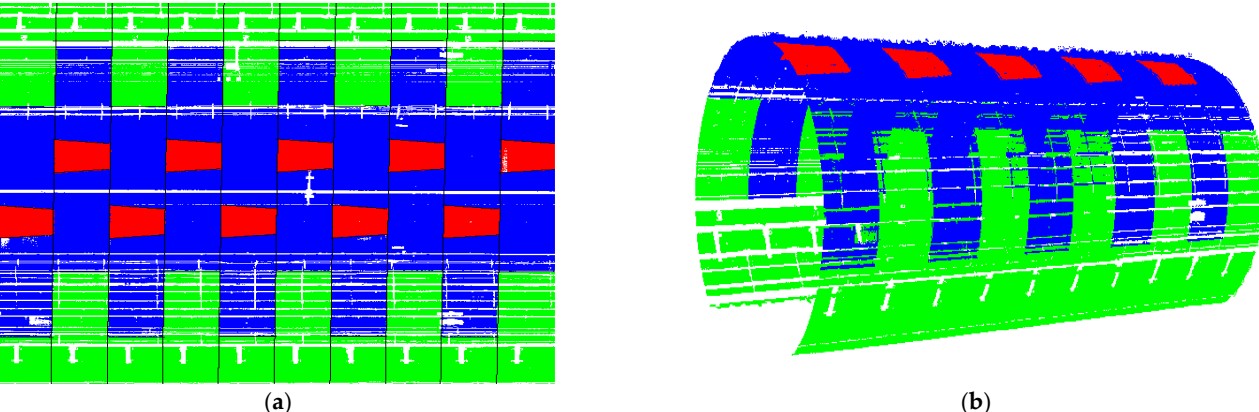

<div align="center">(<b>a</b>)            (<b>b</b>)</div>

**Figure 11.** Positioning effect of a shield tunnel in the (**a**) expanded state and (**b**) original state.

## 3. Results

A circular shield tunnel section of the Tianjin operating subway was selected to verify the effectiveness of the proposed method. The starting mileage of the specific survey area was K19 + 143, length was approximately 100 m, ring width was 1.2 m, inner diameter of tunnel lining was 5.5 m, lining was constructed of precast reinforced concrete segments, and segment splicing method was staggered splicing. The point cloud data from the shield tunnel were collected using the mobile tunnel measurement system developed by Capital Normal University. It consists of an electric inspection vehicle, Z+F 9012 scanner and its battery, laptop computer (or industrial tablet), and acquisition control software. The acquisition control software is deployed on the laptop (or industrial tablet) and connected to the scanner through a local area network.

The Z+F 9012 scanner has three types of resolution (low, high, and ultra-high) and three rotational speeds (50, 100, and 200 Hz). The scanner can simultaneously collect coordinates and laser reflectivity information, and its technical parameters are summarized in Table 2. The electric inspection vehicle can be driven at a constant speed of 0.05–1.25 m/s and adjusted through five gears. The constant speed error is less than 0.5%. The mobile tunnel measurement system based on Z+F 9012 can quickly acquire high-precision, high-density tunnel laser point cloud data. The following parameters were selected for this study: High resolution, rotation speed of 100 Hz, 0.5 m/s for the forward speed of the car to scan and collect along the track, and one section of data was selected for analysis. The experimental field is shown in Figure 12a, and the point cloud data of the shield tunnel laser scanning are shown in Figure 12b, this section of data contains 24,619,547 points.

**Table 2.** Z+F 9012 laser scanner parameters.

| Laser System | |
|---|---|
| Beam divergence | <0.5 mrad |
| Ranging range | 0.3–119 m |
| Ranging resolution | 0.1 mm |
| Measuring point rate | Maximum 1.016 million points per second |
| Linearity error | ≤1 mm |
| **Transmitter Unit** | |
| Vertical viewing angle | 360° |
| Angular resolution | 0.0088° (40,960 pixel/360°) |
| Angle accuracy | 0.02° rms |
| Rotating speed | 50–200 Hz (Highest 12,000 rpm) |

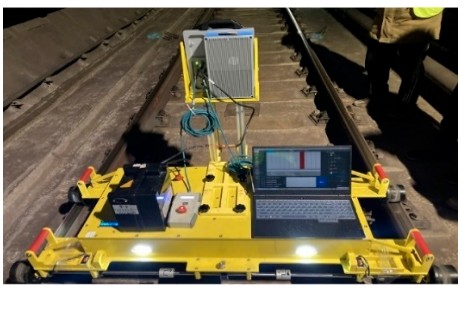

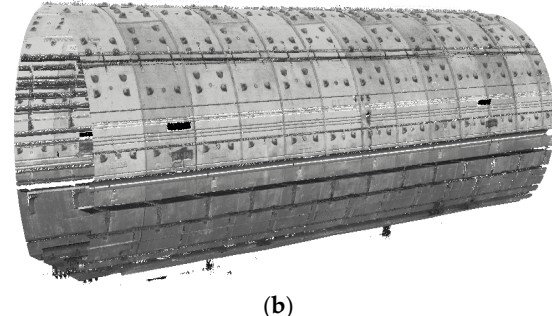

<div align="center">(<b>a</b>)          (<b>b</b>)</div>

**Figure 12.** (**a**) The experimental field and (**b**) laser point cloud data of the shield tunnel.

### 3.1. Experimental Analysis

The results were analyzed based on three evaluation metrics: (1) extraction rate of bolt holes, namely, the percentage of the number of extracted bolt holes from all bolt holes, (2) positioning accuracy of longitudinal seams, and (3) positioning accuracy of circumferential seams. The seam in this study was a straight line. Therefore, the straight line of the seam was first discretized into point clusters with coordinate information at 1 mm intervals. Then, the accuracy of seam location was compared with the seam in the point cloud.

#### 3.1.1. Extraction Rate of Bolt Holes

There were 300 bolt holes in the test data. In addition, there were 298 bolt holes in the original point cloud since two bolt holes were completely covered by emergency exit and vertical curve signs, as shown in Figure 13a. The point clouds of some bolt holes were missing owing to the occlusion of wires in the tunnel, as shown in Figure 13b. These bolt holes did not meet the density required by DBSCAN clustering, and thus were outliers and eliminated. Finally, 291 bolt holes which met the requirements of locating seams according to the distribution characteristics of bolt holes, were extracted using the proposed method with an extraction rate of 97.65%.

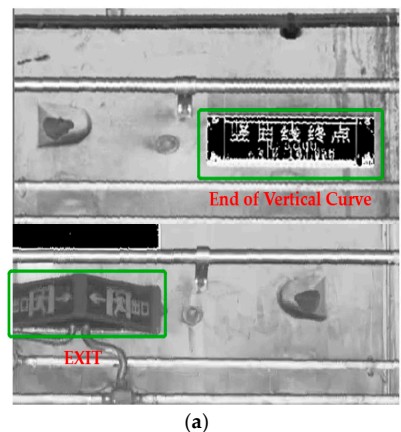

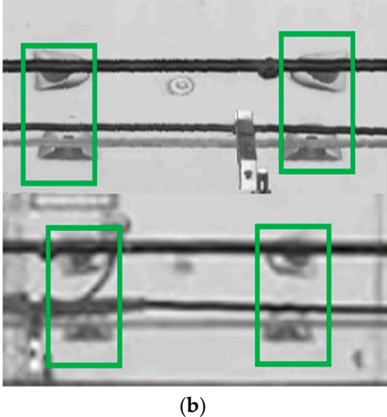

<div align="center">(<b>a</b>)          (<b>b</b>)</div>

**Figure 13.** Shielded bolt holes that were (**a**) completely and (**b**) partially blocked.

#### 3.1.2. Positioning Accuracy of Longitudinal Seams

Four longitudinal seams on both sides between the capping and adjacent blocks and between the adjacent and standard blocks in each ring in the survey area were positioned as shown in Figure 14. The accuracy of the positioning results of the longitudinal seams on both sides only between the capping and adjacent blocks was analyzed since these were unobstructed, whereas other longitudinal seams were blocked by cables.

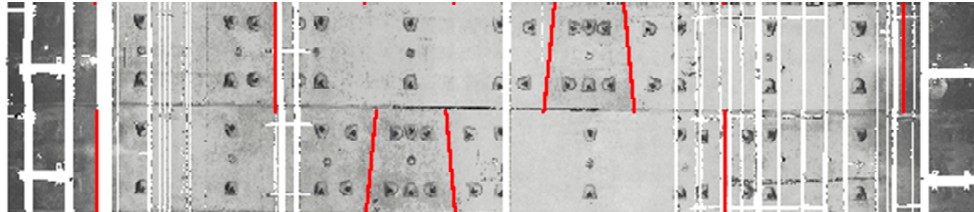

**Figure 14.** Positioning effect of longitudinal seams in the data loop of the survey area.

For the two longitudinal seams formed by the capping block and its left and right adjacent blocks in the same ring, the proposed method is compared with Sun et al.'s method [35], as shown in Figure 15. Black and red lines represent the seam in the point cloud and longitudinal seam located by the proposed method, respectively, as shown in Figure 15a. Green lines represent the longitudinal seam located by the method proposed by Sun et al., as shown in Figure 15b.

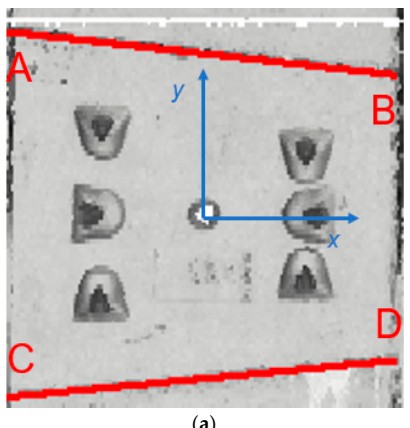
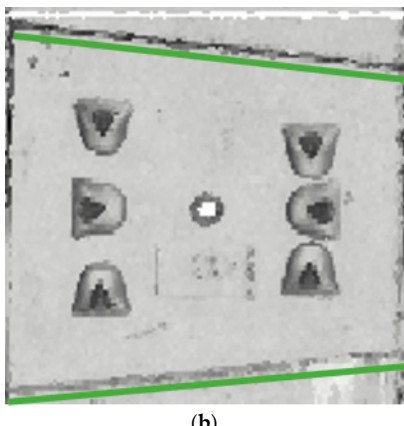

| (a) | (b) |

**Figure 15.** Comparison of longitudinal seams at the same position in the same ring to identify seams of the (**a**) proposed and (**b**) Sun et al.'s method. A, B, C and D are the four corners of the capping block.

The *x*-axis of the coordinate system was parallel to the tunnel centerline and pointed to the direction of large mileage, as shown in Figure 15a. Additionally, the *y*-axis was perpendicular to the *x*-axis. The y-coordinate values of the longitudinal seam in the five-ring point cloud were randomly selected and compared with the A, B, C, and D endpoints of the proposed seam positioning method to evaluate its positioning accuracy. The longitudinal seams in the point cloud were obtained by manual interpretation. The accuracy of manual interpretation can be guaranteed since the color of the longitudinal seams was different from the tunnel wall. The data comparison values of 10 longitudinal seams in the five rings are listed in Table 3.

### 3.1.3. Positioning Accuracy of Circumferential Seam

The mileage value (x-value) was selected to compare the circumferential seams located using the proposed and Du et al.'s method [32] to that in the point cloud at the same position of the vault, left waist, and right waist of the point cloud on the tunnel wall after expansion. This was carried out to analyze the positioning accuracy of the circumferential seam more accurately. The circumferential seams are shown in Figure 16. Black, red, and blue lines are the circumferential seams in the point cloud, positioned by the proposed method, and positioned by Du et al.'s method, respectively.

**Table 3.** Comparison between the original cloud seams and proposed method.

| Serial Number | Location | *Y*-Value of Longitudinal Seam in the Point Cloud (m) | *Y*-Value of Longitudinal Seam Located by this Method (m) | D-Value (mm) |
|---|---|---|---|---|
| 1 | A | 1.552 | 1.554 | 2 |
| | B | 1.449 | 1.443 | 6 |
| | C | 0.497 | 0.495 | 2 |
| | D | 0.606 | 0.599 | 6 |
| 2 | A | −0.568 | −0.560 | 9 |
| | B | −0.675 | −0.677 | 2 |
| | C | −1.624 | −1.617 | 7 |
| | D | −1.507 | −1.512 | 4 |
| 3 | A | 1.550 | 1.552 | 2 |
| | B | 1.433 | 1.429 | 3 |
| | C | 0.482 | 0.482 | 0 |
| | D | 0.590 | 0.588 | 2 |
| 4 | A | −0.569 | −0.567 | 2 |
| | B | −0.689 | −0.691 | 1 |
| | C | −1.630 | −1.635 | 6 |
| | D | −1.530 | −1.529 | 2 |
| 5 | A | 1.528 | 1.530 | 2 |
| | B | 1.423 | 1.419 | 4 |
| | C | 0.468 | 0.463 | 4 |
| | D | 0.579 | 0.578 | 1 |
| Average Difference (mm) | | | | 3.4 |

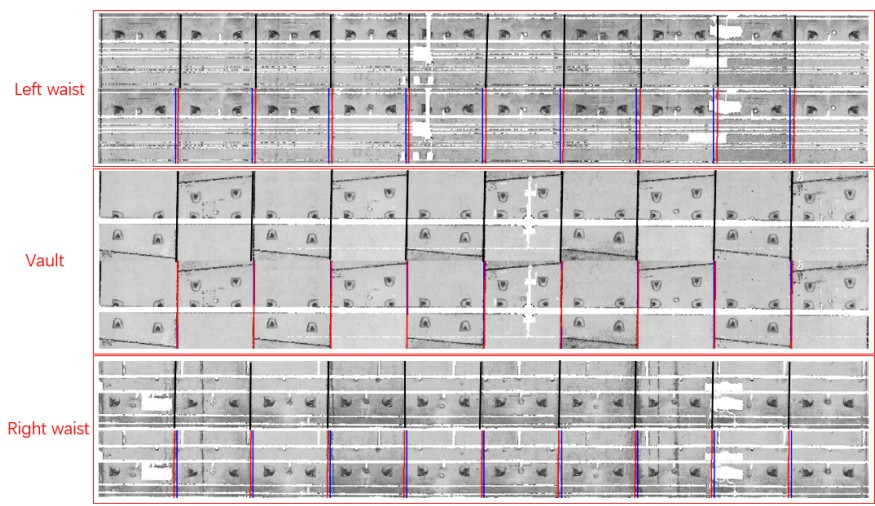

**Figure 16.** Comparison of circumferential seam positioning. Black, red, and blue lines are the circumferential seams in the point cloud, positioned by the proposed method, and positioned by Du et al.'s method, respectively.

The differences between the circumferential seams located by Du et al.'s method, the proposed method, and in the point cloud are listed in Table 4. The circumferential seams in the point cloud were obtained by manual interpretation. The accuracy of manual interpretation can be guaranteed since its color was different from the color of the tunnel wall. The average differences between the circumferential seams located by Du et al.'s method and those in the point cloud at the vault, left waist, and right waist were 5, 51, and 29 mm, respectively. The average differences between the circumferential seams located by

the proposed method and those in the point cloud at the vault, left waist, and right waist were 4, 3, and 5 mm, respectively. The results demonstrated that the proposed method could achieve high-precision localization of the circumferential seams.

**Table 4.** Comparison of the original cloud seams, proposed method, and Du et al.'s method for locating seams.

| Serial Number | Location | Circumferential Seams in the Point Cloud (m) | Circumferential Seams Positioned by Du et al.'s Method (m) | Circumferential Seams Positioned by the Proposed Method (m) | D-Value between Point Cloud and Du et al.'s Method (mm) | D-Value between Point Cloud and the Proposed Method (mm) |
|---|---|---|---|---|---|---|
| 1 | Left waist | 19,144.535 | 19,144.500 | 19,144.537 | 35 | 3 |
| | Vault | 19,144.490 | 19,144.500 | 19,144.497 | 10 | 7 |
| | Right waist | 19,144.462 | 19,144.500 | 19,144.459 | 38 | 3 |
| 2 | Left waist | 19,145.740 | 19,145.690 | 19,145.744 | 50 | 4 |
| | Vault | 19,145.690 | 19,145.690 | 19,145.696 | 0 | 6 |
| | Right waist | 19,145.658 | 19,145.690 | 19,145.651 | 32 | 7 |
| 3 | Left waist | 19,146.940 | 19,146.890 | 19,146.942 | 50 | 3 |
| | Vault | 19,146.895 | 19,146.890 | 19,146.896 | 5 | 1 |
| | Right waist | 19,146.860 | 19,146.890 | 19,146.852 | 30 | 8 |
| 4 | Left waist | 19,148.135 | 19,148.090 | 19,148.141 | 45 | 6 |
| | Vault | 19,148.087 | 19,148.090 | 19,148.092 | 3 | 5 |
| | Right waist | 19,148.058 | 19,148.090 | 19,148.056 | 32 | 2 |
| 5 | Left waist | 19,149.340 | 19,149.290 | 19,149.342 | 50 | 1 |
| | Vault | 19,149.295 | 19,149.290 | 19,149.299 | 5 | 4 |
| | Right waist | 19,149.267 | 19,149.290 | 19,149.258 | 23 | 9 |
| 6 | Left waist | 19,150.554 | 19,150.500 | 19,150.548 | 54 | 6 |
| | Vault | 19,150.495 | 19,150.500 | 19,150.498 | 5 | 3 |
| | Right waist | 19,150.472 | 19,150.500 | 19,150.479 | 28 | 4 |
| 7 | Left waist | 19,151.745 | 19,151.690 | 19,151.746 | 55 | 2 |
| | Vault | 19,151.695 | 19,151.690 | 19,151.697 | 5 | 2 |
| | Right waist | 19,151.660 | 19,151.690 | 19,151.651 | 30 | 9 |
| 8 | Left waist | 19,152.950 | 19,152.890 | 19,152.954 | 60 | 4 |
| | Vault | 19,152.895 | 19,152.890 | 19,152.897 | 5 | 2 |
| | Right waist | 19,152.865 | 19,152.890 | 19,152.867 | 25 | 2 |
| 9 | Left waist | 19,154.149 | 19,154.090 | 19,154.151 | 59 | 2 |
| | Vault | 19,154.095 | 19,154.090 | 19,154.097 | 5 | 3 |
| | Right waist | 19,154.068 | 19,154.090 | 19,154.063 | 22 | 5 |
| Mean value | Left waist | | | | 51 | 3 |
| | Vault | | | | 5 | 4 |
| | Right waist | | | | 29 | 5 |

## 4. Discussion

Aiming at the problem related to the fact that the 3D point cloud positioning of shield tunnel obtained by mobile laser scanning technology is inaccurate, this study proposes a 3D laser point cloud positioning method for shield tunnel, which is divided into rings in the mileage direction and blocks in the ring direction to improve the positioning accuracy of shield tunnel. This study locates the circumferential and longitudinal seams according to the regular spatial distribution law of bolt holes, thus it is very necessary to measure the extraction rate of bolt holes. In the test data, the extraction rate of bolt holes reaches 97.65%, which can meet the requirements of seam positioning. Due to the occlusion of the attachments on the inner wall of the tunnel, the point cloud of some bolt holes is missing, and the DBSCAN algorithm will recognize it as noise and remove it. Circumferential and longitudinal seams are the result of segment splicing of shield tunnel, and are also the key to ring and block division of shield tunnel.

Regarding circumferential seams, some scholars located them by counting the pixel gradient value of the grayscale image within a certain range. The premise of this method was that the circumferential seam in the image generated after projection expansion was

perpendicular to the center line of the track. However, the pipe ring types of the tunnel were divided into standard and wedge rings. A tunnel excavation is a long-term deviation correction process; therefore, it is rare for the centerline of the tunnel to be completely straight. A wedge-shaped ring plays an important role in practical applications, resulting in a certain angle between the ring seam and centerline of the tunnel. Even a standard ring tunnel segment will form a certain angle between the circumferential seam and centerline of the tunnel owing to human factors during construction, which causes the positioning accuracy of the circumferential seam of this method to be low. Given that the arrangement of bolt holes connecting the two rings is parallel to the circumferential seam, this study uses the center point of the bolt hole to fit the straight line to locate the circumferential seams. The test results show that the positioning accuracies of this paper are 1, 48, and 24 mm higher than the traditional method at the vault, left waist, and right waist, respectively.

For the longitudinal seams, some scholars located them on the basis of grayscale images. However, in the actual tunnel, the longitudinal seams will be blocked by tunnel accessories, especially the seams between standard blocks and adjacent blocks that cannot be positioned, and factors such as lighting conditions will also cause inaccurate positioning accuracy. In this paper, the relationship between bolt holes and longitudinal seams is used to locate the longitudinal seams on the basis of 3D point clouds, which avoids situations in which the seams cannot be located and improves the positioning accuracy. In the test data, the longitudinal seams located by this method are compared with those on both sides of the capping block in the original point cloud, and the accuracy difference is 3.4 mm. As the average width of the longitudinal seams in the original point cloud is approximately 5 mm, the positioning accuracy meets the requirements. This study realizes high-precision rings and block division of shield tunnels by high-precision seam positioning. In the future, for bolt holes with missing point clouds owing to shielding, template matching may be used as a method to process missing data, which will be explored in subsequent studies to further improve the accuracy of seam positioning.

## 5. Conclusions

This study proposed a 3D laser point cloud positioning method that was divided into rings in the mileage direction and blocks in the ring direction to improve the positional accuracy for shield tunnels. The traditional data acquisition method is inefficient owing to the limitations of closed tunnel environments and short skylight periods. LiDAR technology is widely used in tunnel engineering due to its advantages, such as high precision, high density, and fast acquisition speed. However, there is inaccurate positioning of the 3D point cloud data of a shield tunnel when obtained by a mobile laser scanning system. This study proposed a method to use the spatial distribution characteristics of bolt holes to locate seams and divided the tunnel into rings and blocks to achieve accurate positioning of the 3D point cloud data of a shield tunnel. A circular shield tunnel section of the Tianjin operating subway was selected to verify the effectiveness of the proposed method. First, the tunnel wall point cloud with bolt holes was obtained using pretreatment methods, such as tunnel track separation, removal of tunnel internal attachments, and projection expansion of the tunnel wall cylindrical surface. Second, an improved CSF algorithm with adaptive parameter adjustment and DBSCAN algorithm were used to extract a bolt hole point cloud. Third, the center points of bolt holes belonging to various segments were obtained using the mean-shift clustering algorithm and bolt hole recognition template. Finally, the center point of the bolt hole was used to fit the straight-line positioning seam to achieve the ring and block division of a shield tunnel.

The accuracy of the positioning seams of the proposed method was compared with the existing methods and seams in the tunnel point cloud through the measured data. The average difference between the longitudinal seams in the point cloud and those positioned by the proposed method was 3.4 mm. The average differences between the circumferential seams positioned using the proposed method and those in the point cloud at the left waist, vault, and right waist were 3, 4, and 5 mm, respectively. The proposed method achieved

high-precision ring and block segmentation of the shield tunnel's 3D laser point cloud, providing important technical and theoretical support for tunnel safety monitoring.

**Author Contributions:** Conceptualization, C.J. and H.S.; methodology, C.J.; software, C.J.; validation, C.J.; formal analysis, C.J. and H.S.; investigation, C.J., J.L. and Y.H.; resources, C.J. and H.S.; data curation, C.J., J.L. and Y.H.; writing—original draft preparation, C.J.; writing—review and editing, C.J. and H.S.; visualization, J.L. and Y.H.; supervision, H.S. and R.Z.; project administration, H.S.; funding acquisition, H.S. and R.Z. All authors have read and agreed to the published version of the manuscript.

**Funding:** This work was supported by National Natural Science Foundation of China under Grant numbers 42101444 and 42071444, in part by the General scientific research projects of Beijing Municipal Commission of Education under Grant number KM202010028012, and Open fund of State Key Laboratory of Rail Transit Engineering Informatization (SKLK22-09).

**Conflicts of Interest:** The authors declare no conflict of interest.

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
