# Peer review of "Precise Positioning Method of Moving Laser Point Cloud in Shield Tunnel Based on Bolt Hole Extraction"

_remotesensing, doi:10.3390/rs14194791_

Round 1

Reviewer 1 Report

Corrections are given in the PDF document.

Author Response

Thank the reviewer for these precious comments concerning my manuscript entitled” Precise Positioning Method of Moving Laser Point Cloud in Shield Tunnel Based on Bolt Hole Extraction”, These comments are all valuable and very helpful for revising and improving our paper, as well as the important guiding significance to our research. We have studied comments carefully and have made corrections which we hope meet with approval. I have uploaded the reply to the reviewer's comments to the attachment.

Reviewer 2 Report

This paper proposed a 3D laser point cloud positioning method to improve the positional accuracy for shield tunnels.  The topic is interesting and matches well for the MDPI Remote Sensing journal.

Some parts of the paper need to be modified and improved:

1, The descriptions of Du's method and "the Canny operator edge detection and Hough transform line detection algorithms" in the introduction are too detailed. Although both methods are used in the experimental section to compare with the proposed method, the description of the review of previous work should be made more concise and clear in the introduction. Also, the traditional methods presented in the introduction seem to be processed on flat images, and point cloud-based processing should also be reviewed.

2, The discussion of the experimental results is insufficient. Although section IV is titled Discussion and Conclusion, there is basically no analysis and discussion of the experimental part. Combining discussion and conclusion into one section is usually an unwise approach and can make it difficult for readers to understand the validity of the proposed method. An additional discussion section is needed to analyze and discuss the overall results of the experiments to enhance the persuasiveness of the paper.

3, Add the number of points of the shield tunnel to the depiction in Figure 12b. The text in line 457 should be in a uniform font.

Author Response

(The authors gave the same response as above.)

Reviewer 3 Report

This paper presents a moving three-dimensional laser point cloud positioning method that is divided into rings in the mileage direction and blocks in the ring direction to improve the positional accuracy for shield tunnels. The research method of this paper is novel, the research technical route is relatively complete, and the research results are credible. The research is of great application value. However, there are still a lots that need to be added and improved, such as:

1. In part 1, the relevant literatures are about gray image and its applications, and it is suggested to supplement the research literatures based on point cloud data.

2. In line 395, Figure 12 missing(a).

3. In line 508-510,” Additionally, the accuracy of the algorithm could 508 be affected if the integrity is defective, such as when the bolt hole was blocked. Addition-509 ally, if the integrity is defective, such as the bolt hole was blocked, the accuracy of the 510 algorithm could be affected.”, is it repeated?

Author Response

(The authors gave the same response as above.)

Round 2

Reviewer 2 Report

I think the revised manuscript can be published in the Remote Sensing journal. Also, when describing the number of points in a large-scale point cloud data, it is common to use scientific notation or separate the numbers with spacers, for example, in line 384 "24619547 points" could be expressed as "2.46 x 107 points" or "24,619,547 points".